# Evolution of Gut Microbiome and Metabolome in Suspected Necrotizing Enterocolitis: A Case-Control Study

**DOI:** 10.3390/jcm9072278

**Published:** 2020-07-17

**Authors:** Camille Brehin, Damien Dubois, Odile Dicky, Sophie Breinig, Eric Oswald, Matteo Serino

**Affiliations:** 1General Pediatrics Department, Children Hospital, Toulouse University Hospital, 31300 Toulouse, France; brehin.c@chu-toulouse.fr; 2IRSD, Université de Toulouse, INSERM, INRAE, ENVT, UPS, 31024 Toulouse, France; dubois.d@chu-toulouse.fr (D.D.); eric.oswald@inserm.fr (E.O.); 3Service of Bacteriology-Hygiene, Toulouse University Hospital, 31300 Toulouse, France; 4Department of Neonatology, Children Hospital, University Hospital, UMR 1027, INSERM, Paul Sabatier University, 31000 Toulouse, France; dicky.o@chu-toulouse.fr; 5Neonatal and Pediatric Intensive Care Unit, Toulouse University Hospital, 31300 Toulouse, France; breinig.s@chu-toulouse.fr

**Keywords:** necrotizing enterocolitis, intestinal microbiology, microbiome, infant gut, metabolomics

## Abstract

Background: Necrotizing enterocolitis (NEC) is a devastating condition in preterm infants due to multiple factors, including gut microbiota dysbiosis. NEC development is poorly understood, due to the focus on severe NEC (NEC-2/3). Methods: We studied the gut microbiota, microbiome and metabolome of children with suspected NEC (NEC-1). Results: NEC-1 gut microbiota had a higher abundance of the Streptococcus (second 10-days of life) and Staphylococcus (third 10-days of life) species. NEC-1 children showed a microbiome evolution in the third 10-days of life being the most divergent, and were associated with a different metabolomic signature than in healthy children. The NEC-1 microbiome had increased glycosaminoglycan degradation and lysosome activity by the first 10-days of life, and was more sensitive to childbirth, low birth weight and gestational age, than healthy microbiome. NEC-1 fecal metabolome was more divergent by the second month of life. Conclusions: NEC-1 gut microbiota and microbiome modifications appear more distinguishable by the third 10-days of life, compared to healthy children. These data identify a precise window of time (i.e., the third 10-days of life) and provide microbial targets to fight/blunt NEC-1 progression.

## 1. Introduction

Necrotizing enterocolitis (NEC), defined by the Bell classification [1,2,3], is the most severe intestinal disease in preterm infants, with a mortality score of 25% and long-term neurological morbidity [4]. Yet, a precise initiating factor of this pathology is missing. In the last decade, gut microbiota was identified and recognized as a specific organ with functions widely beyond digestion [5]. Both its taxonomic (relative abundance) and functional (microbial pathway) alterations, named dysbiosis, were described in several pathologies, in particular, metabolic diseases such as type 2 diabetes and obesity [6,7,8], and intestinal inflammatory diseases [9,10]. Importantly, a dysbiotic gut microbiota associated with a very high inflammatory status of the gut [11,12] may trigger NEC development, since germ-free mice do not develop NEC [13].

From a clinical and microbiological point of view, studies of NEC were focused only on established and severe phenotypes, such as NEC-2 and NEC-3. Based on the French study Etude épidémiologique sur les petits âges gestationnels (EPIPAGE 2), the incidence of proved NEC-2 and NEC-3 is 1–5% in preterm infants born at less than 32 weeks of gestation [14].

By contrast, NEC suspicions such as lethargy, bradycardia, thermic instability associated with biliary gastric residues, vomiting, abdominal distension with or without rectal bleeding, with a normal abdominal x-ray image or a simple dilatation, which identifies suspected NEC (NEC-1), have not been studied yet. In fact, enteropathies are frequent in the first weeks of life in preterm infants, though no data are available about NEC-1 incidence. This induces the end of alimentation, a prolonged (sometime life-lasting) parenteral nutrition, with a delayed gut maturation and failure to thrive [15]. Therefore, to study the evolution of gut microbiota and microbiome during the early onset of NEC, we focused on NEC-1 children within the first two months of life. We studied the fecal metabolome, to understand how a change in gut microbiota may drive alterations in intestinal metabolites. To further understand which factor of mother and child may affect the evolution of gut microbiota, microbiome and fecal metabolome during NEC-1, we analyzed: the presence of neonatal antibiotherapy (ABx), ABx treatment on the mother, childbirth (Cesarean-section (C-sec) vs. vaginal birth (VB)), very low birth weight (VLBW), extreme low birth weight (ELBW) and gestational age (GA) > or ≤ 28 weeks.

## 2. Research Design and Methods

### 2.1. Cohort Constitution

We conducted a prospective monocentric case-control cohort study. This study was approved (number of the approval: DC 2016-2804) by Neonatal and Pediatric Intensive Care Unit and Neonatology Department of Purpan Hospital in Toulouse, France. The parents of the children involved in this study gave their approval by written consent. The inclusion criteria regarding all of the children hospitalized into the Neonatal and Pediatric Intensive Care Unit or Neonatology Departments of the Purpan Hospital, were:− Newborn of gestational age under 34 weeks of gestation;− Diagnosis of suspected necrotizing enterocolitis (NEC-1) made by a neonatologist;− Obtainment of the non-opposition from parents or their legal representative.

Following the inclusion of every case, we conducted in parallel a search for two controls, according to the following matching criteria, listed in decreasing priority:− Gestational age (±1 week of gestation, priority to matched age);− Body weight;− Neonatal antibiotherapy; − Childbirth (C-section vs. vaginal);− Maternal antibiotherapy. 

Inclusion criteria for controls were:− Newborn of gestational age under 34 weeks of gestation;− Respect of the matching according to the priority order of the established criteria; − Obtainment of the oral non-opposition from parents or their legal representative.

Children with complex congenital cardiopathy or with spontaneous intestinal perforation without a radiological evidence of NEC were excluded from the study.

Based on these criteria, we included 11 NEC-1 children, with 27 feces collection (4 fecal samples for time-point 1–10 days (d); 10 fecal samples for time-point 11–20 d; 7 fecal samples for time-point 21–30 d; 6 fecal samples for time-point > 30 d) and 21 healthy children, with 53 feces collection (15 fecal samples for time-point 1–10 d; 14 fecal samples for time-point 11–20 d; 13 fecal samples for time-point 21–30 d; 11 fecal samples for time-point > 30 d). Hence, a total of 80 fecal samples was analyzed in our study. The period of collection was day 1 to day 68 of life of the newborn.

### 2.2. Taxonomic and Functional Analysis of Gut Microbiota

Feces analyzed in this study were collected by nurses in the related department in the first week of life and once a week till the end of the hospitalization. Feces were firstly kept at 4 °C in a 5 mL Eppendorf tube with 20% glycerol/Lysogeny Broth and then stored at −80 °C. Total DNA was extracted from feces as previously described [16], with a modification: a thermic shock of 30 seconds was performed between each bead-shaking step (3 bead-shaking steps of 30 seconds each at maximum speed). The 16S bacterial DNA V3–V4 regions were targeted by 357wf-785R primers and analyzed by MiSeq (RTLGenomics, Texas, USA). An average of 68,669 sequences was generated per sample. Bioinformatic filters applied as already described [17]. Cladogram and LDA scores were drawn via the LEfSe algorithm [18]. Diversity indices were calculated using the software Past 4.02 (Hammer, Ø., Harper, D.A.T., and P. D. Ryan, 2001. PAST: Paleontological Statistics Software Package for Education and Data Analysis. Palaeontologia Electronica 4(1): 9pp). The predictive functional analysis of the gut microbiota was performed via PICRUSt [19]. Diseases and host genetic variation linked to NEC-1_21–30 d-associated gut microbiota were identified via MicrobiomeAnalyst [20], with the Taxon Set Enrichment Analysis module.

### 2.3. Fecal Metabolome Analysis

The metabolome (total metabolites) analysis of the feces was performed as previously described [17]. Fecal samples have been prepared as it follows: 50 mg of feces were homogenized for 30 seconds in 500 µL of a pH 7.0 phosphate buffer, prepared in D_2_O. Then, the homogenate was chilled into ice for 1 minute and centrifuged at 12,000 RPM for 10 minutes at 4 °C. The supernatant was then recovered, and the pellet was re-homogenized again at the same conditions of 12,000 RPM for 10 minutes at 4 °C. All the supernatants were then pooled and centrifuged at 18,000 RPM for 30 minutes at 4 °C. The supernatant was recovered and centrifuged again at the same conditions of 18,000 RPM for 30 minutes at 4 °C. A total of 600 µL of the final supernatant was then analyzed into nuclear magnetic resonance (NMR) tubes of 5 mm of diameter. The conformity criterium to validate the final sample was the aspect, to be crystal clear.

Pathway-associated metabolite sets and SNP-associated metabolite sets (Appendix A) were analyzed via MetaboAnalyst 4.0 [21], with the enrichment analysis module.

### 2.4. Statistical Analysis

The results are presented as mean ± SEM for histograms and box and whiskers graphs. Statistical analyses were performed by two-way analysis of variance (ANOVA) followed by a two-stage linear step-up procedure of Benjamini, Krieger and Yekutieli to correct for multiple comparisons by controlling the false discovery rate (<0.05) (for histograms) or the Mann-Whitney test (for box and whiskers), as indicated in the figure legend, by using GraphPad Prism version 7.05 for Windows Vista (GraphPad Software, San Diego, CA, USA). For Table 1, results are presented as median or as indicated and *P* value was calculated using Fisher’s exact test. Significant values were considered starting at *P* < 0.05. For the taxonomical and predictive functional analysis of gut microbiota, significant values were considered, starting at *P* < 0.05 or *P* < 0.01 when indicated. Principal component analysis (PCA) graphs were drawn by using Past 4.02.

## 3. Results

### 3.1. Analysis of Gut Microbiota, Microbiome and Fecal Metabolome During NEC-1

To understand the microbial and metabolomic evolution during the early onset of necrotizing enterocolitis (NEC), we studied clinical profile suspected NEC (NEC-1) preterm infants. NEC-1 children underwent more glycopeptides treatment, showed significantly higher cordon lactates, bacteremia and a longer full enteral feeding, when compared to age-matched healthy children (Table 1). NEC-1 children also displayed a lower plasma pH and enteral milk volume at day 7 (Appendix A) and a higher abundance of *Streptoccoccus* species (Appendix A) compared to healthy children. Both populations of children showed a high intragroup variance in terms of gut microbiota (Appendix A) and overall microbial diversity (Appendix A). NEC-1 microbiome showed increased activity for pathway related to transcription, glycosaminoglycan degradation and lysosome, compared to healthy children (Appendix A). Then, we analyzed the fecal metabolome to appreciate NEC-1-induced changes in gut microbial metabolic activity. NEC-1 children displayed a reduced intragroup variation and significantly lower levels of ethanol (Appendix A). Overall, these data show that NEC-1 is characterized by a precise gut microbiota, microbiome and gut microbial metabolites profile.

### 3.2. Analysis of Gut Microbiota, Microbiome and Fecal Metabolome During the Evolution of NEC-1 over the First Two Months of Life

Given the presence of a NEC-1-specific gut microbiota and microbiome profile, we aimed to identify at what time these profiles establish. We divided both NEC-1 and healthy children populations in subgroups according to periods of ten days of life as it follows: 1–10 d (d stands for “days”), 11–20 d, 21–30 d for the first month of life and > 30 d for the second one. In the first 10 days, NEC-1 children displayed a divergent and more homogenous gut microbiota compared to healthy children, with the latter characterized by a higher abundance of *Klebsiella* species (Figure 1A,B). At this stage of life, gut microbiota in NEC-1 had a lower diversity based on Chao-1 index (Figure 1C) and a different microbial activity related to replication, recombination and repair proteins, lysosome and glycosaminoglycan degradation (Figure 1D). No significant changes were observed in fecal metabolites (Figure 1E). Overall, these data show that gut microbiome starts to diverge at the early onset of NEC-1.

In the second 10-days of life, NEC-1 gut microbiota was characterized again by a higher abundance of *Streptococcus* species and bacteria from the Micrococcales order (Figure 2A), with a high intragroup variance (Figure 2B). At this stage of life, NEC-1 gut microbiota also showed a higher diversity based on Chao-1 index (Figure 2C), but no microbial pathway differently regulated (Figure 2D). As for the fecal metabolome, NEC-1 children displayed significant lower levels of serine (Figure 2E). Overall, these data show a stronger evolution of gut microbiota than gut microbiome in the second 10-days of life, between NEC-1 and healthy children.

In the third 10-days of life, changes in NEC-1 gut microbiota compared to healthy children occurred to a bigger extent and were related to increased *Staphylococcus* and *Streptococcus* species (Figure 3A,B), together with a high intragroup variance (Figure 3C) and no change in the overall diversity indices (Figure 3D). We also observed a NEC-1 microbiome profile, mainly based on thiamine and seleno-compound metabolism (Figure 3E). The NEC-1 gut microbiota profile of the third 10-days of life was associated with: (i) multiple diseases and found significantly increased in ulcerative colitis (Figure 4A); (ii) host genetic variation and significantly related to ANP32E, a gene involved in ulcerative colitis [22], in line with previous reports. In terms of fecal metabolome, we observed no significant changes in NEC-1 vs. healthy children (Figure 4C). Then, we studied feces collected in the second month of life. In this period of life, the taxonomical differences in the gut microbiota of NEC-1 vs. healthy children were related to the increase in *Raoultella* species in NEC-1 gut microbiota (Figure 5A), with a still high intragroup variance (Figure 5B), and no change in the overall microbial diversity indices (Figure 5C). We also observed microbial functions related to DNA repair increased in the NEC-1 gut microbiome (Figure 5D). This period of life was characterized by the highest separation in terms of fecal metabolome, with significant lower levels of ethanol and leucine in NEC-1 children (Figure 5E).

### 3.3. Specific Impact of NEC-1 on the Evolution of Gut Microbiota. Microbiome and Fecal Metabolome over the First Two Months of Life, Compared to Healthy Children

To investigate the evolution of gut microbiota, microbiome and fecal metabolome over the first two months of life, we conducted an intra-group study in both NEC-1 and healthy children, according to the four groups reported above: 1–10 d, 11–20 d, 21–30 d and > 30 d. We did not observe any taxonomic significant change in the gut microbiota of NEC-1 children. However, the group NEC-1_21–30 d had a specific gut microbiome with an increased restriction enzyme activity, among others (Appendix A). The four NEC-1 groups also differed in terms of fecal metabolome, with regard to leucine, ethanol and serine amounts (Appendix A). Based on these results, we performed a metabolomic enrichment analysis on two levels: (i) pathway-associated metabolite sets (Appendix A) and (ii) single nucleotide polymorphism (SNP)-associated metabolite sets (Appendix A). NEC-1 metabolomic profile (increased ethanol and serine) was significantly associated with both homocysteine degradation and phosphatidylethanolamine biosynthesis (Appendix A), with serine being the metabolite the most linked to NEC-1-associated SNP (Appendix A). By contrast, in healthy children, the four groups reported above did not differ in terms of both gut microbiota and microbiome, but only with regard to fecal metabolome (Appendix A). Healthy metabolomic profile (increased leucine, ethanol and dihydroxyacetone) was significantly associated with valine, leucine and isoleucine degradation and to ketone body metabolism (Appendix A), with leucine being the metabolite the most linked to healthy-associated SNP (Appendix A). Overall, these data suggest that: (i) a different intragroup evolution exists between NEC-1 and healthy children, with regard to gut microbiota and microbiome; and (ii) the NEC-1 microbiome appears to be more sensitive to mother-related factors.

### 3.4. Maternal and Child Factors Influencing the Gut Microbiota, Microbiome and Fecal Metabolome During NEC-1

Next, we asked which factor related to both mother and child may affect the most the above reported parameters. We analyzed six conditions: neonatal antibiotherapy (ABx), ABx treatment on mother, childbirth (C-section (C-sec) vs. vaginal birth (VB)), very low birth weight (VLBW), extreme low birth weight (ELBW) and gestational age (GA) > or ≤ 28 weeks.

Only neonatal ABx treatment affected the gut microbiota in both NEC-1 and healthy children (Appendix A). By contrast, all the above factors, except the VLBW, affected the gut microbiome (Appendix A). Note that childbirth modality, ELBW and GA affected the gut microbiome only in NEC-1 children (Appendix A). Moreover, all the above factors, except the neonatal ABx treatment and ELBW, affected the fecal metabolome between NEC-1 and healthy children (Appendix A). Then, we performed a metabolomic enrichment analysis on the pathway-associated metabolite sets, based on Appendix A, in which there is an increase in ethanol and succinate within in the NEC-1_GA ≤ 28 w. Ketone body and butyrate metabolism were the most significantly associated with this metabolomic set (Appendix A).

## 4. Discussion

In this prospective study, we focused on suspected necrotizing enterocolitis NEC-1 preterm infants. NEC-1 phenotype has been poorly clinically investigated, with no data available on gut microbiota, microbiome and fecal metabolome. By contrast, NEC-2 and NEC-3, more severe and established phenotypes, have been more characterized. As for clinical parameters, the increased cordon lactate levels we found in NEC-1 has been recently positively correlated to the development of enteropathy [23]. Hence, the hypothesis of hypoxic lesions in utero or during birth may not be excluded, and could even be predictive of neonatal morbidity. Importantly, the observed reduced enteral nutrition volume in NEC-1 is not a protective factor during the development of severe NEC (NEC-2 and NEC-3), but rather, it may lengthen hospitalization and infections risk [24]. Moreover, a recent multicentric study showed that a slow rate enteral feeding is associated with an increased risk of developing NEC-2 and NEC-3 [14].

NEC-1 children showed a high general variance for gut microbiota and fecal metabolome, which is in line with a personalized microbiota and fecal metabolome profiles of preterm infant [25]. This is evident from PCA analyses in Figure 4 C and Figure 5 E, where the metabolome profiles of the healthy vs. NEC-1 children appear to diverge, but are still presenting some overlap, due to high intragroup variance. Both this evidence and the delayed intestinal colonization of preterm infants [26,27] may explain the lack of NEC-1-specific microbial group in the first 10-days of life. The analysis by periods of ten days of life revealed a divergence for both gut microbiota and microbiome in NEC-1 by the third 10-days of life. In particular, the higher abundance of *Staphylococcus* in NEC-1 is in accordance with the early colonization by *Staphylococcus* bacteria of the intestine of preterm infants [28]. This datum suggests the third 10-days as an optimal time window to be targeted by antibiotics directed against bacterial species higher in NEC-1, such as *Staphylococcus*. However, in our study NEC-1 children who underwent glycopeptide and aminoglycoside therapy were more numerous than healthy children. Therefore, this evidence suggests that NEC-1 may be associated with glycopeptide and/or aminoglycoside-resistance, since NEC-1 gut microbiota was characterized by an increase, and not a decrease, of *Staphylococcus.* Since aminoglycosides are active antibiotics against enterobacteria, their administration could delay intestinal colonization by Proteobacteria, and thus promote the implantation of resistant genera, such as *Staphylococcus* and *Streptococcus*. Based on this evidence, our data suggest not to prolong antibiotic therapy beyond the first week of life in preterm infants, if inflammatory parameters have been normalized. Furthermore, NEC-1 gut microbiota profile was associated with ulcerative colitis and host genetic variation in the ANP32E gene, encoding a protein implicated in cortico-resistance during ulcerative colitis [22]. NEC-1 children showed increased exposition to antenatal corticosteroids compared to healthy children, even though a study has not identified antenatal corticosteroids as a NEC-inducing factor [29]. Despite Anp32e-deficient mice display no sign of disease [30], it has not to be excluded the role of Anp32e in a model of gut inflammation mimicking ulcerative colitis. Hence, further studies are warranted on genetic factors of NEC. In terms of microbial functions, the intragroup analysis showed in the third 10-days of life a higher restriction enzyme activity in the NEC-1 gut microbiome. This bacterial activity, directed against bacteriophages and enriched in the newborn intestine [31], suggests an increased virus activity and, hence, a virome dysbiosis, beyond a microbiota dysbiosis, during NEC-1 evolution. Notably, with regard to the time-point > 30 days, microbiota and microbiome changes are still present and, most importantly, different from those observed at earlier time-points. This suggests that NEC-1 onset may have influenced the evolution of gut microbiota. All these microbial data are associated with our observation about a change in fecal amino-acids, such as leucine and serine, confirming the association between gut microbiota dysbiosis and a change in amino-acids metabolism [32].

Finally, several factors may affect the gut microbiota of preterm infants, e.g., hospital regimens, even with regard to the putative use of probiotics to blunt NEC. In a very recent publication, Kurath-Koller et al. showed the efficacy of some probiotics in improving the gut microbiome of very low birth weight infants during the first two weeks of age, in a triple-center cohort study [33]. In our mono-centric study, the very low birth weight had an impact on the fecal metabolome of both healthy and NEC-1 children, but not on their gut microbiota nor on their microbiome. This evidence underlines the importance of the hospital environment for the evolution of both an eubiotic and a dysbiotic, e.g., during NEC-1, gut microbiota.

## 5. Conclusions

The small cohort represents a limitation of our work, due to the difficulty to investigate NEC at its early onset. However, our data may provide neonatal departments with immediate indications to blunt NEC-1 evolution such as: i) increasing the enteral volume of nutrition, especially in the first days of life; ii) revising and reducing antibiotic therapy up to the first week of life in preterm infants.

Availability of data and materials: All data are available in the main text or the supplementary materials and via the following repositories: Sequence Read Archive (SRA) database with the assigned identifier PRJNA579480.

Ethics approval and consent to participate: This study was approved (number of the approval: DC 2016-2804) by Neonatal and Pediatric Intensive Care Unit and Neonatology Department of Purpan Hospital in Toulouse, France. The parents of the children involved in this study gave their approval by written consensus.

## Figures and Tables

**Figure 1 jcm-09-02278-f001:**
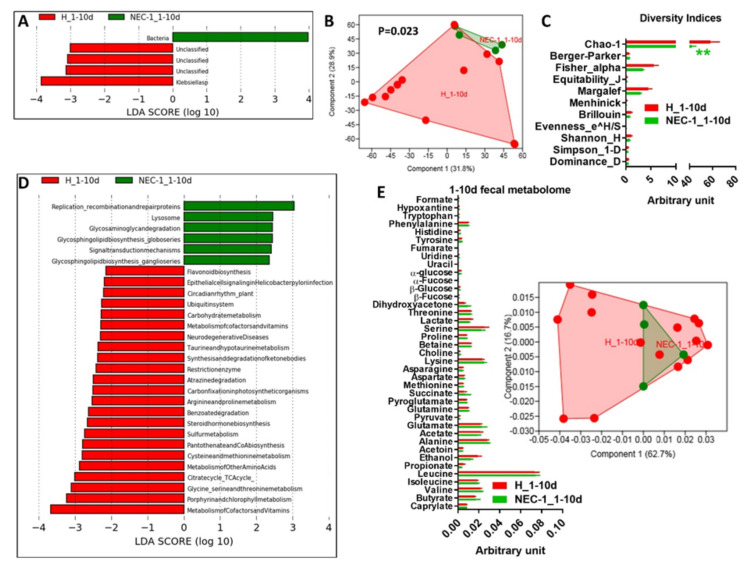
Analysis of gut microbiota, microbiome and metabolome in the first 10-days of life in healthy vs. necrotizing enterocolitis (NEC)-1 children. (**A**) Gut microbiota analysis via linear discriminant analysis (LDA) score between healthy (H) vs. NEC-1 children, in the first 10-days of life, 1 to 10 days; (**B**) principal component analysis (PCA) of the gut microbiota; (**C**) indices of gut microbiota diversity; (**D**) LDA score for microbial pathways; (**E**) histogram of the overall fecal metabolites and PCA as inset. ***P* < 0.01. Two-way ANOVA, followed by a two-stage linear step-up procedure of Benjamini, Krieger and Yekutieli to correct for multiple comparisons, by controlling the false discovery rate (<0.05); *N* = 15 for H and *N* = 4 for NEC-1.

**Figure 2 jcm-09-02278-f002:**
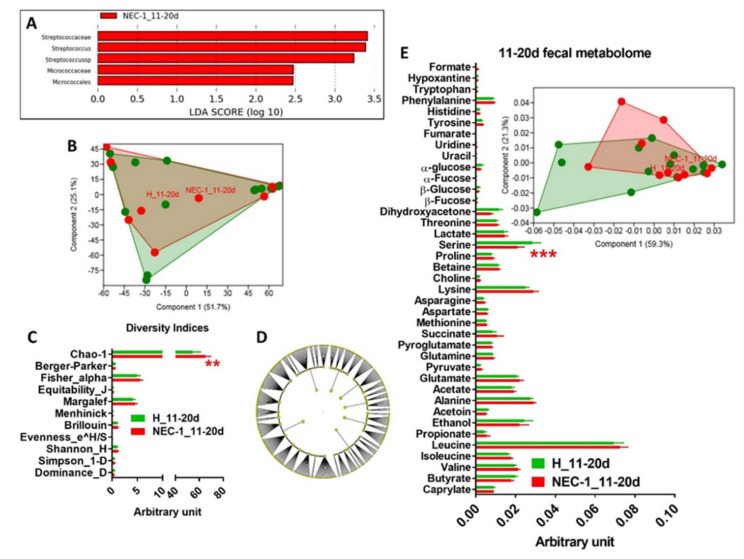
Analysis of gut microbiota, microbiome and metabolome in the second 10-days of life in healthy vs. NEC-1 children. (**A**) Gut microbiota analysis via LDA score between healthy (H) vs. NEC-1 children, in the second 10-days of life, 11 to 20 days (d) (the LDA score is only shown for NEC-1 children meaning that no bacteria are significantly higher in the H group vs. NEC-1); (**B**) PCA of the gut microbiota; (**C**) indices of gut microbiota diversity; (**D**) null cladogram for microbial pathways; (**E**) histogram of the overall fecal metabolites and PCA as inset. ***P* < 0.01. ****P* < 0.001. Two-way ANOVA, followed by a two-stage linear step-up procedure of Benjamini, Krieger and Yekutieli to correct for multiple comparisons by controlling the false discovery rate (<0.05); *N* = 14 for H and *N* = 10 for NEC-1.

**Figure 3 jcm-09-02278-f003:**
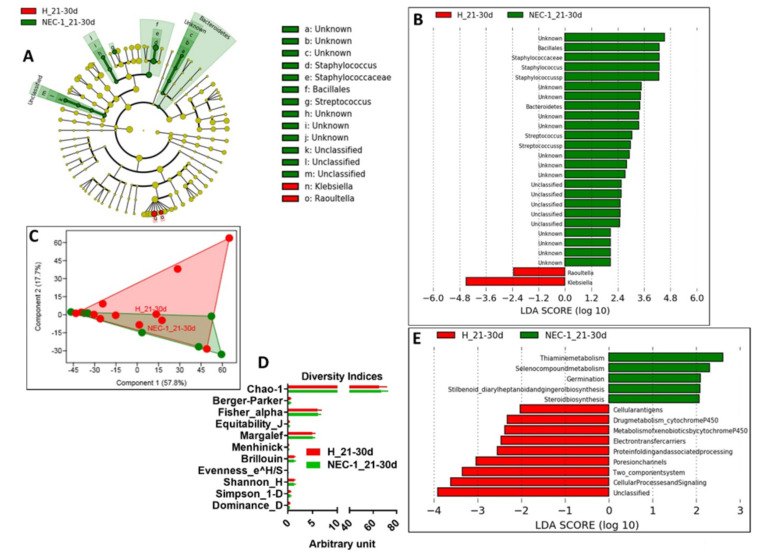
A specific gut microbiota and microbiome exist in the third 10-days of life in healthy vs. NEC-1 children. (**A**) Comparative analysis of the gut microbiota by LDA effect size (LEfSe): the cladogram shows bacterial taxa significantly higher in the group of children of the same color, in the fecal microbiota between healthy (H) vs. NEC-1 children, in the third 10-days of life, 21 to 30 days (**D**) (the cladogram shows the taxonomic levels represented by rings with phyla at the innermost and genera at the outermost ring and each circle is a bacterial member within that level); (**B**) LDA score used to build the cladogram in (A); (**C**) PCA of the gut microbiota; (**D**) indices of gut microbiota diversity; (**E**) LDA score for microbial pathways. *N* = 13 for H and *N* = 7 for NEC-1.

**Figure 4 jcm-09-02278-f004:**
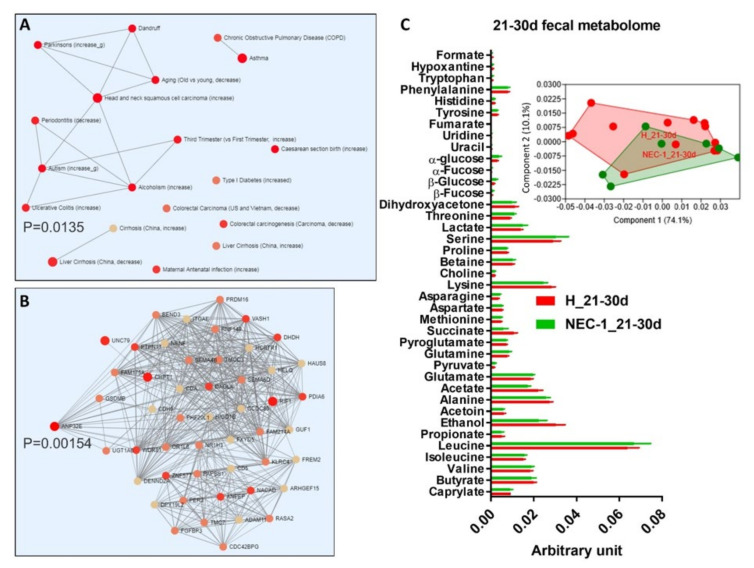
Diseases, host genetic variation and metabolome analysis in the third 10-days of life during NEC-1. (**A**) Diseases and (**B**) host genetic variation linked to NEC-1_21–30d associated gut microbiota; (**C**) histogram of the overall fecal metabolites and PCA, as inset. *N* = 13 for H and *N* = 7 for NEC-1.

**Figure 5 jcm-09-02278-f005:**
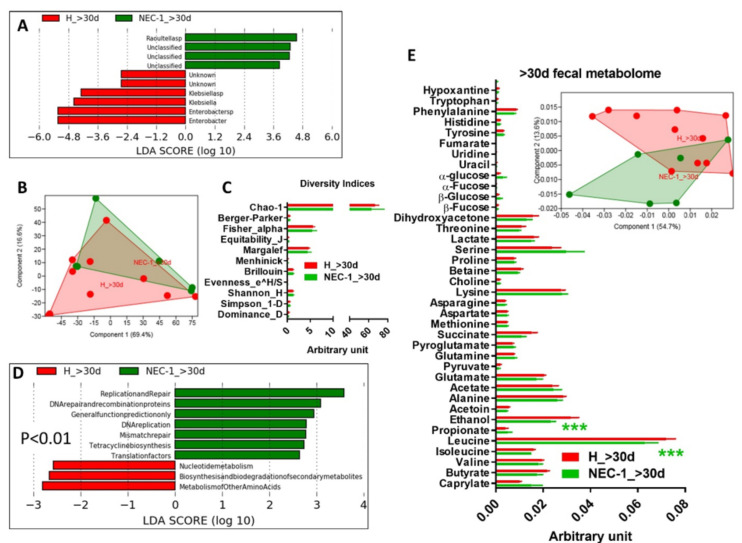
Analysis of gut microbiota, microbiome and metabolome in the second month of life in healthy vs. NEC-1 children. (**A**) Gut microbiota analysis via LDA score between healthy (H) vs. NEC-1 children, in the second month of life > 30 days (d); (**B**) principal component analysis (PCA) of the gut microbiota; (**C**) indices of gut microbiota diversity; (**D**) LDA score for predictive microbial pathways (*P* < 0.01); (**E**) histogram of the overall fecal metabolites and PCA, as inset. ****P* < 0.001. Two-way ANOVA, followed by a two-stage linear step-up procedure of Benjamini, Krieger and Yekutiel,i to correct for multiple comparisons by controlling the false discovery rate (<0.05); *N* = 11 for H and *N* = 6 for NEC-1.

**Table 1 jcm-09-02278-t001:** Cohorts characteristics.

Variables, Description	NEC-1 *n* = 11	Healthy *n* = 21	*P* (Fisher’s Exact Test)
Birth weight, median, inter-quartile (IQ)	1150 (845–1815)	1360 (700–2105)	Not significant (ns) (> 0.05)
Gestational age, median (weeks) (IQ)	28.4 (26–31)	30 (26, 4–32)	ns
Gender Girls, number (%) Boys, number	2 (18) 9	8 (38) 13	ns
Patent Ductus arteriosus, number (%)	3 (27)	6 (28)	ns
Parity, number (%)	3 (27)	3 (10)	ns
Antenatal corticosteroids, number (%)	11 (100)	19 (90)	ns
Hypertension, eclampsia, number (%)	2 (18)	3 (14)	ns
Multiple births, number (%)	2 (18)	4 (19)	ns
Antenatal antibiotics, number (%)	4 (36)	5 (24)	ns
Chorioamniotitis, number (%)	2 (18)	1 (5)	ns
Apgar Score, 1 min (IQ) 5 min (IQ)	8 (1–10) 10 (4–10)	7 (1–10) 8 (1–10)	ns ns
Cordon pH (IQ)	7.23 (6.8–7.4)	7.31 (7.04–7.43)	ns
Cordon lactates (IQ)	5.7 (2.9–9.6)	3 (1–7.2)	0.04 * (<0.05)
Mean arterial pressure at hospital admission (IQ)	29 (20–47)	29.5 (17–43)	ns
Hospital Admission T (°C) (IQ)	36.5 (35.1–37.8)	36.8 (35.8–37.5)	ns
Antibiotics in the first week of life (%)	10 (90)	18 (85)	ns
Days under antibiotics (IQ)	7.5 (2–17)	3 (0–18)	ns
Days under antibiotics (Third-Generation-Cephalosporin (3GC) ± Penicillin A, ±aminoglycoside) in the first week of life (IQ)	3 (0–7)	3 (0–7)	ns
Children under glycopeptides number (%)	8 (72)	3 (14)	0.0018 ** (<0.01)
Bacteremia, number (%)	5 (45)	1 (5)	0.01 * (<0.05)
Exposition to mother milk, number (%)	11 (100)	21 (100)	ns
Age of enteropathy (days) (IQ)	12 (4–60)	-	-
Exposition to inotropes, number (%)	1 (9)	0 (0)	ns
Blood transfusion, number (%)	2 (18)	4 (19)	ns
Full enteral feeding (days) (IQ)	23 (14–39)	11 (3–29)	0.0002 *** (<0.001)
Median fasting time (median in days) (IQ)	1 (1–4)	-	-

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
