# Peer review of "Evolution of Gut Microbiome and Metabolome in Suspected Necrotizing Enterocolitis: A Case-Control Study"

_jcm, 2020, doi:10.3390/jcm9072278_

Round 1

Reviewer 1 Report

Dear authors,

regarding the manuscript under evaluation, although it is very interesting and rich in content, it is dealing with some issues. In my opinion the introduction is broad and would greatly benefit from a review of previous related studies.

As far as the experimental design is concerned, the groups of infants that were enrolled in the study and the sample collection are a little confusing (lines 86-88). A graphical illustration would be of great help to the reader. The part of fecal metabolome analysis, although it is of great significance to the paper, it is not described in detail, although many steps of the metabolomics based workflow could affect the obtained data.

Another small issue is the way the results are presented. Separating the metabolome from the microbiome part in two different paragraphs would help the reader have a better understanding of the content of the manuscript. For example, a more extensive discussion about PCA of figures 4 and 5 could be beneficial to the manuscript.  

Finally, more emphasis should be given in the discussion part. For example, a comparison of the findings of the microbiome and the metabolome part with previous studies would greatly improve the paper. Furthermore, a conclusion should be added about the correlation of Staphylococcus and Streptococcus induced metabolome alteration with the pathophysiology of NEC, essentially connecting lines 291 and 292 with the manuscript’s findings.

Yours sincerely

Reviewer 2 Report

The manuscript “Evolution of Gut Microbiome and Metabolome in Suspected Necrotizing Enterocolitis: a Case-Control Study” by Brehim et al. describes the microbiome in premature infants with feeding intolerance. The clinical relevance of these observations is unclear.

  1. The clinical relevance of this study and of NEC-1 is unclear. The duration of feeding intolerance in the patients is not clearly stated. In our unit, we treat NEC-1 with 2 days of antibiotics and 3 days of fasting, just to observe these infants closely and hopefully find out at an early stage if any of these patients progress to more severe stages of NEC. If none of the patients showed clinical progression, the concern is unclear. Are the authors suggesting that infants with a peculiar colonization pattern had longer or recurrent feeding intolerance?
  2. The number of infants with NEC-1 is small (n=11). Because infants with NEC-1 do not have a unifying clinical, laboratory, or imaging sign, we do not even know if these infants belonged to a homogeneous group of patients. In the absence of any unifying diagnostic criteria for NEC-1, the scientific/clinical value of these data is uncertain. These infants could have had multiple disorders, or the feeding intolerance could have just been due to immaturity. In premature infants, transient feeding intolerance is common. Transient lethargy, bradycardia, thermic instability, biliary gastric residues, vomiting, abdominal distension, with normal abdominal x-ray images are frequently seen in these patients.
  3. It seems that none of these infants progressed to more severe grades of NEC. The association of this microbiome with a specific clinical profile would have been useful.
  4. Infants with NEC-1 had been fed longer, and so gastrointestinal function may not even have been a concern. Did they have altered outcomes such as impaired growth, higher mortality, or longer length of stay? Unfortunately, the extremely small number of patients makes it difficult to draw any meaningful inferences.
  5. Was there a difference if the number of days or the proportion of parenteral nutrition?

Reviewer 3 Report

Authors studied the gut microbiota, microbiome and metabolome of children with suspected NEC (NEC-1) and found that NEC-1 gut 22 microbiota had a higher abundance of Streptococcus (second 10-days of life) and Staphylococcus 23 (third 10-days of life) species. NEC-1 children showed a microbiome evolution in the third 10-days of life being the most divergent and associated to a different metabolomic signature than in healthy children. NEC-1 microbiome had increased glycosaminoglycan degradation and lysosome activity by the first 10-days of life and was more sensitive to childbirth, low birth weight and gestational age, than healthy microbiome. NEC-1 fecal metabolome was more divergent by the second month of life. Conclusions: NEC-1 gut microbiota and microbiome modifications appear more distinguishable by the third 10-days of life, compared to healthy children. These data identify a precise window of time (i.e. third 10-days of life) and provide microbial targets to fight/blunt NEC-1 progression.

There are several major concerns to mention

Methods are questionable as far as abdominal signs and symptoms might be associated with developing NEC  or in a majority of cases are associated with divergent entities, thus what do we learn from these preterm infants and their microbiota evaluation. How can the authors talk about a precise window of time between days 21 and 30 of life (in order to do what? with the result of what?)?

There are no infants having progressed to definite diagnosis of NEC ≥IIa, thus, again the same question as above.

Subgroup investigation on microbiome development >2 months of age is of low value. The findings do not explain any hypothesis postulated and correlate with…?

There are a lot of findings that were described in all details. For example: If you make venipuncture for blood picture you can describe all the possible findings “in extenso” and in case of different methods for analysis lots of sentences might be written, but shortly WBC and hematocrit and platelets are within normal range or not, and that`s it.

“Based on this evidence, our data suggest not to prolong antibiotic therapy beyond the first week of life in preterm infants.” I do not agree with the authors. What is the idea of antibiotics for 1 week? Hopefully: early-onset sepsis. What is the idea of prolonged antibiotic courses: to eliminate pathogens with an antibiotic that does not fit exactly (pathogen unknown) with probably slowly normalizing inflammatory indices. Nobody would stop antibiotics at day 7 if CRP values are still abnormal!? It is not possible to postulate such a sentence based on different microbiome findings which have to be discussed and interpreted carefully. And in case of prophylactic antibiotics this message by the authors is still well known – irrespective of microbiome analysis. And I do not agree with the conclusion that higher volumes regarding enteral feeding might be one conclusion, and neonatologists are advised to do so. Regarding abdominal signs and symptoms associated with NEC-1, nobody would increase enteral volumes.

There are a lot of factors influencing the preterm gut microbiome e.g. Kurath-Koller et al,  Nutrients 2020 that have to be considered during analysis of  the gut microbiome.

Therefor conclusions are not based on study findings

Minor concerns

Table 1 would be easier/better to read if given ns = not significant (if p>0,05) instead of senseless numbers like >0.9999

Round 2

Reviewer 1 Report

Regarding the revised manuscript, although the authors gave some satisfactory responses to reviewer’s comments, no extended alteration in the text occurred.  However, this does not change the fact that this is an interesting work that could be published regardless.

Reviewer 3 Report

The manuscript improved substantially. No further major or minor concerns.